# Prognosis of Advanced Heart Failure Patients according to Their Hemodynamic Profile Based on the Modified Forrester Classification

**DOI:** 10.3390/jcm11133663

**Published:** 2022-06-24

**Authors:** Guillaume Baudry, Juliette Bourdin, Raluca Mocan, Elisabeth Hugon-Vallet, Matteo Pozzi, Antoine Jobbé-Duval, Nicolas Paulo, Patrick Rossignol, Laurent Sebbag, Nicolas Girerd

**Affiliations:** 1Service d’insuffisance Cardiaque, Hôpital Cardiovasculaire Louis Pradel, 69500 Bron, France; juliette.bourdin@chu-lyon.fr (J.B.); ralucapisau@yahoo.com (R.M.); elisabeth.hugon-vallet@chu-lyon.fr (E.H.-V.); antoine.jobbe-duval@chu-lyon.fr (A.J.-D.); nicolas.paulo@chu-lyon.fr (N.P.); laurent.sebbag@chu-lyon.fr (L.S.); 2Centre d’Investigations Cliniques Plurithématique 1433, INSERM DCAC, CHRU de Nancy, F-CRIN INI-CRCT, Université de Lorraine, 54500 Vandoeuvre-lès-Nancy, France; p.rossignol@chru-nancy.fr (P.R.); n.girerd@chru-nancy.fr (N.G.); 3Service de Chirurgie Cardiaque, Hôpital Cardiovasculaire Louis Pradel, 69500 Bron, France; matteo.pozzi@chu-lyon.fr

**Keywords:** advanced heart failure, Forrester classification, systemic congestion, acute kidney injury, acute heat failure, heart transplantation waiting list

## Abstract

Introduction: Heart transplantation (HT) remains the gold-standard treatment but is conditioned by organ shortage. This study aimed to evaluate the value of Forrester classification and determine which congestion criteria had the best prognostic value to predict cardiorenal events on heart transplant waiting list. Methods and results: One hundred consecutive patients (54 years old, 72% men) with available right heart catheterization (RHC) listed in our center for HT between 2014 and 2019 were included. Cardiac catheterization measurements were obtained at the time of HT listing evaluation. Patients were classified according to perfusion and congestion status in four groups: “warm and dry”, “warm and wet”, “cold and dry”, and “cold and wet”. pWet was used to classify patients with pulmonary congestion and sWet for systemic congestion. The primary endpoint was the rate of a composite criteria of cardiogenic shock, acute kidney injury, and acute heart failure. Secondary endpoint was the incidence of waitlist death, emergency HT, or left ventricular assist device (LVAD) implantation at 12 months evaluated by Kaplan–Meier curves and log-rank test. Only Forrester classification according to systemic congestion was associated with the primary composite endpoint (*p* = 0.011), while patients’ profile according to pulmonary congestion was not (*p* = 0.331). Similarly, only the Forrester classification according to systemic congestion predicted waitlist death, emergency HT, or LVAD implantation at 12 months, with *p* = 0.010 and *p* = 0.189 for systemic and pulmonary congestion, respectively. Moreover, systemic congestion was the main driver of cardiorenal events on waitlist. Conclusions: Forrester classification according to systemic congestion is associated with cardiorenal outcomes in patients listed for heart transplant and the risk of waitlist death, emergency HT, or LVAD implantation at 12 months.

## 1. Introduction

Heart failure (HF) is a public health problem affecting 1% to 2% of the European population, and advanced or refractory heart failure affects 1% to 10% of this population [1,2]. In the last decades, advanced heart failure population has grown parallelly with the increase of heart failure patients due to the effectiveness of therapeutics, allowing a prolonged survival. Heart transplant remains the gold-standard therapy for refractory heart failure. However, access is limited by organ shortage. Indeed, only 61% of listed patients are transplanted at 6 months, and 12% died or were delisted for worsening in 2019 during the same time frame [3]. Thus, it is necessary to identify patients on the waitlist with adverse prognosis. Clinical assessment of perfusion (warm or cold) and congestion (dry or wet) is recommended by the ESC for patients with acute HF since 2016 [4]. This recommendation came from Forrester’s classification, published in 1978 by James Forrester [5], who classified patients in four groups (warm and dry, warm and wet, cold and dry, cold and wet) after myocardial infarction. This classification has a double interest: assess severity and adapt treatment. Patients with advanced heart failure experience episodes of low perfusion and congestion; however, the interest of Forrester’s classification is this population remain unclear. Right heart catheterization (RHC) is necessary to assess hemodynamic parameters, especially pulmonary vascular resistance of heart transplant candidates [4,6], and allow assessment of Forrester’s profiles. Congestion during RHC could be defined as systemic or pulmonary according to central venous pressure (CVP) or pulmonary capillary wedge pressure (PCWP), respectively. 

This study aimed to evaluate the value of Forrester classification to predict cardio-renal events while listed for heart transplant and assess whether systemic or pulmonary congestion criteria has the best prognostic value. 

## 2. Materials and Methods

### 2.1. Study Population

This study consisted of a mono-center cohort analysis using the French national CRISTAL database. All 201 new adult patients (18 years or older) registered on the French heart transplant waiting list between 1 January 2014 and 31 December 2019 at the Hospices Civiles de Lyon heart transplant center were included. In order to assess the value of right heart catheterization on events, patients without a cardiac catheterization and patients that were on the HT list for less than 30 days were excluded. 

### 2.2. Pulmonary Artery Catheterization

Right heart catheterization measurements were obtained as part of the assessment for heart transplant listing according to the 2016 International Society for Heart Lung Transplantation guidelines (Class 1, Level of Evidence: C). Elevated pulmonary vascular resistance (PVR) that is refractory to medical therapy may be a contraindication to HT [6]. The most recent catheterization was used in those having undergone multiple exams before listing.

Hemodynamic variables obtained during cardiac catheterization included systolic blood pressure (SBP), diastolic blood pressure (DBP), cardiac output (CO), pulmonary capillary wedge pressure, systolic pulmonary arterial pressure (sPAP), diastolic pulmonary arterial pressure (dPAP), mean pulmonary arterial pressure (mPAP), and right atrial pressure (RAP) as indicator of central (CVP). Cardiac index (CI) is determined from cardiac output divided by body surface area (BSA), calculated from the Dubois formula (BSA = 0.007184 × Size^0.725^ × Weight^0.425^). The PAPi (pulmonary artery pressure index) corresponded to (sPAP-dPAP)/CVP. The RVSWi (right ventricular stroke work index) corresponded to CI × mPAP × 0.0144.

### 2.3. Data Collection

This single-center cohort was performed using the data of the French national transplant registry (CRISTAL). This registry is managed by the Agence de la biomédecine and prospectively collected data from every candidate for organ transplant in France [7]. Data collection is required by the transplant agency. Data such as demographic parameters, device therapy, mechanical circulatory support, or occurrence of death were collected from CRISTAL. Cardiac events (acute heart failure, cardiogenic shock, acute kidney injury) while waiting for HT, biological parameters, and treatments at the time of cardiac catheterization were retrospectively collected. 

### 2.4. Definition of Events

“Acute heart failure” was defined as an unscheduled hospitalization for management of an episode of congestive heart failure with introduction of intravenous diuretics.

“Cardiogenic shock” was defined by a hospitalization in intensive care unit requiring intravenous inotropic support.

“Acute kidney injury” was defined according to KDIGO [8] (Kidney Disease: Improving Global Outcomes) criteria: stage 1—increase plasma creatinine ≥ 26.5 μmol/L or 1.5 to 1.9 times baseline plasma creatinine; stage 2—2.0 to 2.9 times baseline plasma creatinine; and stage 3—3.0 times baseline plasma creatinine or plasma creatinine ≥ 354µmol/L or initiation of renal replacement therapy.

Analysis of the occurrence of the composite end point of emergency heart transplantation, emergency mechanical assistance, or death is assessed from the cardiac catheterization.

Estimated glomerular filtration rate was estimated using the CKDepi formula [9] (eGFR = 141 × min(CR/k, 1)^α^ × max(CR/k, 1)^−1.209^ × 0.993^age^ × 1.018 (for female patients) × 1.159 (for black patients), which α: −0.329 (female) or −0.411 (male) and k: 61.9 (female) or 79.6 (male)).

The study was conducted according to French legislation indicating that identified that research studies based on the national CRISTAL registry do not require institutional review board approval.

### 2.5. Modified Forrester Classification

Using cardiac catheterization measures of CO, PCWP, and CVP (assessed from RAP), the study population was classified into four groups according to hemodynamic state (estimated by CI—“warm” or “cold”) and pulmonary congestion (estimated by PCWP—“dry” or “wet”) as Forrester classification. In parallel, the study population was classified according to hemodynamic state and systemic congestion (estimated by CVP).

Cold patients were defined by a cardiac index ≤ 2.0 L/min/m²; pWet was used to classify patients with pulmonary congestion defined by pulmonary capillary wedge pressure ≥ 20 mmHg and sWet for systemic congestion defined by central venous pressure ≥ 10 mmHg.

### 2.6. Primary and Secondary Endpoints

The primary composite endpoint was the annual rate of cardiogenic shock, acute kidney injury, and acute heart failure. Rates were the mean of the number of events during the waiting period on heart transplantation list divided by the time spent on waitlist in years in each patient and were expressed in events/patient*year.

Secondary endpoint was one-year waiting list death or urgent heart transplant or LVAD implantation evaluated by Kaplan–Meier curves and log-rank test.

### 2.7. Statistical Analysis

Baseline continuous variables were expressed as medians with interquartile 25–75% (IQ_25–75_) range or mean (± standard deviation). Comparisons of continuous variables were carried out using Kruskal–Wallis test and *t*-test as required. Categorical variables were expressed as frequencies (percentages) and compared using chi-square test. Comparisons of events rate (acute heart failure, cardiogenic shock, AKI, and global events) were carried out using Kruskal–Wallis test. The two-tailed significance level was set at *p* < 0.05. Outcomes were reported according to Forrester’s groups. Secondary endpoint was assessed and illustrated by Kaplan–Meier analyses using log-rank test. After graphically verifying the assumption of proportional hazard model, multivariable Cox proportional hazard regression was used to examine the association between Forrester’s profile and outcome. A multivariable model adjusted for baseline characteristics (age, gender) and eGFR was built. Correlation between PCWP and CVP was assessed using Pearson’s correlation test.

All analyses were performed using SPSS version 27 (IBM SPSS Statistics for Windows, Armonk, NY, USA: IBM Corp). 

## 3. Results

### 3.1. Baseline Characteristics

Among the 100 patients included, 72% were men, and the median age was 54 years. They received guideline directed treatments (more than 80% received beta-blockers, angiotensin-converting enzyme (ACE) inhibitors, or an angiotensin-2 receptor blockers or combination of valsartan-sacubitril and mineralocorticoid receptor antagonists). A minority of patients had comorbidities (12% diabetes, 22% hypertension, and 15% had an eGFR inferior to 60 mL/min/1.73 m²). All patients had severe left ventricular dysfunction with a median LVEF of 25% and were severely symptomatic (mean NYHA 2.76 and mean VO2max 11.2 mL/kg) (Table 1a,b).

Whether based on PCWP or CVP, eGFR was significantly different between group (*p* = 0.002 according to PCWP and *p* = 0.013 according to CVP). Loop diuretic dose was significantly different between Forrester’s profile according to PCWP (*p* = 0.010) but only tended to differ when CVP was considered (*p* = 0.058). Other treatments were comparable between groups except ARNI: Patients with systemic congestion had a lower likelihood of receiving ARNI (*p* = 0.019).

### 3.2. Primary Endpoint

#### 3.2.1. Analysis According to Pulmonary Congestion

The rate of primary criteria was 2.04, 5.36, 5.66, and 5.47 for “Warm and pDry”, “Warm and pWet”, “Cold and pDry”, and “Cold and pWet” profiles, respectively (*p* = 0.331). There was also no difference between the four groups in the rate of acute heart failure (*p* = 0.594), cardiogenic shock (*p* = 0.430), or acute kidney injury (*p* = 0.404) (Figure 1A).

#### 3.2.2. Analysis According to Systemic Congestion

The rate of primary criteria was 1.74, 7.43, 4.91, and 6.59 for “Warm and sDry”, “Warm and sWet”, “Cold and sDry”, and “Cold and sWet” profiles, respectively (*p* = 0.011). “Cold and sWet” (*p* = 0.007) and “Warm and sWet” (*p* = 0.009) had a higher rate of the primary criteria compared with “Warm and sDry” profiles. Other intergroup comparisons were not significant despite a trend of increased rate for “Cold and sWet” (*p* = 0.075) and “Warm and sWet” (*p* = 0.065) profiles compared with “Cold and sDry” profile. Similarly, the rate of AKI was different among groups, 0.57, 3.31, 1.92, and 2.45 for “Warm and sDry”, “Warm and sWet”, “Cold and sDry”, and “Cold and sWet” profiles, respectively (*p* = 0.049) (Figure 1B). Intergroup comparison showed a significant difference between “Warm and sWet” and “Warm and sDry” profiles (*p* = 0.010) (Figure 1B).

### 3.3. Secondary Endpoint

#### 3.3.1. Analysis According to Pulmonary Congestion

The percentage of waitlist death, emergency HT, or LVAD implantation at 12 months was 36.0%, 26.3%, 22.2%, and 13.2% in “Cold and pWet”, “Warm and pWet”, “Cold and pDry”, and “Warm and pDry” profiles, respectively (*p* = 0.189) (Figure 2A). Forrester’s profile according to pulmonary congestion were not significantly associated with the risk of one-year waitlist death, urgent heart transplant, or LVAD implantation in multivariate analysis (Appendix A).

#### 3.3.2. Analysis According to Systemic Congestion

The percentage of waitlist death, emergency HT,, or LVAD implantation at 12 months was 47.6%, 30.8%, 16.1% and 14.3% in “Cold and& sWet”, “Warm and sWet”, “Cold and sDry”, and “Warm and sDry” profiles, respectively (*p* = 0.010) (Figure 2B). The “Cold-Wet” profile was associated with a 240% increase in the risk of one-year waitlist death, urgent heart transplant, or LVAD implantation while adjusting for age, gender, and eGFR at RHC (HR, 3.40, 95%CI, 1.12 to 10.29; *p* = 0.030) (Appendix A).

#### 3.3.3. Mortality on Waitlist

Ten patients died on the waiting list (six patients related to cardiomyopathy, two from a pathology unrelated to heart disease, and two of unknown cause).

### 3.4. Correlation between PCWP and CVP

Correlation between PCWP and CVP was good, and Pearson correlation was 0.490 (*p* = 0.01) (Figure 3). Correlation was good for patients without pulmonary congestion; they were free from systemic congestion in 82.7% and 77.8% for “Warm” and “Cold” profiles, respectively. On the other hand, correlation was weaker when patients had pulmonary congestion. Indeed, in pulmonary congestive patients, systemic congestion was found only in 53% of the “Warm” profile and 40% of the “Cold” profile.

## 4. Discussion

In this retrospective study, we showed that Forrester’s classification is feasible and useful to stratify the risk of cardiorenal events for patients awaiting heart transplantation. Moreover, Forrester’s classification using systemic congestion (instead of pulmonary congestion) predicted the rate of composite criteria of acute heart failure, cardiogenic shock, acute kidney injury, and the risk of waitlist death, urgent LVAD implantation, or heart transplantation.

### 4.1. Interest of Forrester Classification for Predicting Cardiorenal Outcomes

Described in the late 1970s, Forrester’s classification was later set aside to reappear in the early 2000s. Indeed, whether by clinical [10] or invasive [11] assessment, several studies demonstrated its prognostic and therapeutic interest in acute heart failure. Clinical evaluation of patient profile is recommended in acute heart failure by the ESC 2016 guidelines [4] to assess prognosis and optimize treatment. Interestingly, Chioncel et al. [12] confirmed the prognostic value of this classification at admission and discharge of patients after acute heart failure hospitalization. The prognostic value of this classification in advanced heart failure population independently of acute episodes is still a subject of debate. Nohria et al. [10] showed in 2003 the prognostic values of Forrester’s classification according to clinical assessment in a population of advanced HF patients. However, in this study, the majority of patients were decompensated and elective transplant evaluation for chronic HF accounted for only 17%. Moreover, the main difference with our study was congestion assessment, which was defined by left and right heart failure signs (recent history of orthopnea and/or physical exam evidence of jugular venous distention, rales, hepatojugular reflux, ascites, peripheral edema, leftward radiation of the pulmonic heart sound). More recently, Ibe et al. [13] showed the interest of Forrester’s classification in a heterogenous population including 44% of midrange or preserved HF patients. Generalization of previous published results in our population of patients awaiting HT was difficult for two main reasons: First, our population of advanced HF patients had a median EF of 25% and was more than 10 years younger. Second, cardiac index cut off was 2.5 L/min/m² in the Ibe et al. study compared with 2.0 L/min/m² in our study (which was also the mean CI), highlighting the difference in severity between the two populations. The main finding of our study is to prove the value of Forrester’s classification for predicting cardiorenal endpoint including cardiogenic shock, congestive HF decompensation, and acute kidney injury in addition to a composite of waitlist death, urgent LAVD implantation, or heart transplantation. To the best of our knowledge, it is the first association to a composite cardiorenal endpoint with hemodynamic profile classification.

### 4.2. Congestion Is the Main Driver of Cardiorenal Events on Heart Transplant Waitlist

Systemic congestion (assessed by RAP) was associated with poor prognosis in our study and seems to be the main driver of events on HT waiting list. It is indeed in accordance with literature, there is a growing amount of data on the role of congestion in heart failure. In the study published by Chioncel et al. [12], congestive patients had an increased one-year mortality whether congestion was assessed clinically at admission or discharge. Chronic congestion in HF patients, defined by persistent congestion at discharge or congestion in outpatients, is related to adverse prognosis [14,15]. More recently, a tailored lung ultrasound-guided therapy showed a reduction in a composite of urgent visits, hospitalization for worsening HF, and death, mainly resulting from a decrease in the number of urgent visits for worsening HF [16].

However, no association was found between cold profiles and primary cardiorenal endpoint in our population of advanced HF patients. The rate of cardiogenic shock, acute heart failure, and acute kidney injury did not differ significantly between groups. It is concordant with a recent meta-analysis [17], including most studies published more than 20 years ago, that found no association between low cardiac output and all-cause mortality in advanced HF population awaiting HT. 

Congestion seems to be also the main driver of cardiorenal events. Indeed, Damman et al. [18] highlighted in patients with a wide range of cardiopathies the association between increased CVP and impaired renal function and the independent association with all-cause mortality. Association between increased CVP and impaired renal function was confirmed specifically in a population of patients admitted for acute HF in the Escape trial [19]. Similarly, our team showed that advanced HF patients listed for HT with CVP >10 mmHg doubled the likelihood of eGFR < 60 mL/min/1.73 m² after adjustment for age, history of hypertension, and BNP Z-score, and we found a correlation of eGFR and CVP on a continuous scale [20]. The association between AKI and elevated central venous pressure in acute decompensated HF was reported by Mullens et al. [21] in 2009. Interestingly, patients who developed WRF had a greater baseline CI, and change in GFR was similar between those with CI above and below the mean admission CI, suggesting no CI impact on WRF. Moreover, patients with higher AKI stages had also higher baseline right atrial pressure. Cardiac output was not different between patients developing AKI (regardless of stage) and those without. It has also been described that systemic congestion in advanced HF patients awaiting HT was associated with postoperative AKI [22].

### 4.3. Systemic Congestion Has Higher Prognostic Value to Predict Cardiorenal Events on HT Waitlist

We hypothesized and showed that systemic and pulmonary congestion did not have the same prognostic value to identify cardiorenal events in patients awaiting heart transplantation. Despite a good correlation in our study and literature between RAP and PCWP [23,24,25], this association is weaker among patients with high filling pressures, especially when measurements were close to the defined cut-off. In the advanced heart failure population awaiting heart transplantation, hemodynamic data (CVP and PCWP) were non-concordant in about 25%. In our study, 30% of “Dry” patients according to CVP (cut-off of 10 mmHg) were classified “Wet” according to PCWP (cut-off of 20 mmHg). Congestion assessment by a CVP elevation was more specific than assessment by PCWP and could explain the difference in prognostic value of the two indices. 

PCWP was neither associated with baseline GFR in patients hospitalized for acute heart failure nor worsening renal function [19,21]. In patients listed for HT, our team previously demonstrated the absence of association between PCWP and GFR at time of listing, and this study confirmed that there is no association with AKI while on the waitlist [20].

Surprisingly, an association with one-year waitlist death, urgent HT, or LVAD implantation was found only with systemic congestion in our cohort. Interestingly, this is concordant with a recent study of Sokolska et al. [26] on patients admitted for acute decompensated heart failure. Contrary to isolated pulmonary congestion, presence of systemic congestion on admission was an independent predictor for all-cause mortality within 1 year. One explanation could be the lack of power. Indeed, despite the unfavorable prognosis being associated with pulmonary congestion [14,27], isolated pulmonary congestion is associated with less-frequent in-hospital heart failure worsening during the first 48 h, shorter length of hospital stay, and lower one-year all-cause mortality compared with isolated systemic or mixed (systemic and pulmonary) congestion. This finding was in concordance with the paper of Damman et al. [28] showing that mortality rates almost doubled from no signs to three or more signs of congestion, arguing for an additional deleterious effect of pulmonary and systemic congestion. Pellicori et al. [14] evaluated ultrasound indices of pulmonary and systemic congestion. They found that NTproBNP value increased with increased number of congestion indices. Pulmonary congestion assessed by B-lines was associated with all-cause mortality or heart failure hospitalization in univariate but not in multivariate analysis, contrary to signs of systemic congestion (inferior vena cava diameter or decreasing jugular vein diameter before and after a Valsalva maneuver).

### 4.4. Clinical Implications

First, we showed that patients listed for heart transplantation had a high rate of cardiorenal events while on the waitlist. Our data reinforced idea that congestion, especially systemic congestion, is the main driver of cardiorenal events in patients with advanced heart failure. There in emerging literature on available tools to evaluate congestion. These tools are listed in a recent expert review [29] and could be divided in clinical score, circulating biomarkers, imaging markers, or pressure- and impedance-based tools. Bedside inferior vena cava or lung ultrasound are easy, noninvasive, and rapid methods to verify decongestion before discharge and during heart failure visits [30] and should be systematic in advanced heart failure patients. Decongestion should be the treatment goal whether patients are ambulatory or hospitalized and whether they are symptomatic or not. Observance to loop diuretic must be reinforced. Indeed, loop diuretic omission is associated with immediate decreased of natriuresis and urine output and a risk of heart failure imbalance [31].

Secondly, acute kidney injury accounts for almost one-third of events on the waitlist, and particular attention should be paid to this complication. AKI is a life-threatening situation leading to a short-term risk of death, secondary to hyperkaliemia and hemodynamic compromise, but also long-term risk of chronic kidney disease and death [32,33]. AKI and hyperkaliemia are common causes of temporary or definitive down titration or discontinuation of mineralocorticoid receptor antagonists and RAAS blockers, while these two therapeutic classes are prognostic in HF and after AKI [34,35]. We agree with Ronco et al. [36] arguing that there is a need for specialists that are experts in the pathophysiology and the clinical manifestations of cardio-renal syndrome in order to prevent and correctly treat AKI in context of advanced HF. Moreover, elevated rate of AKI suggests that addition of AKI criteria at primary outcome in heart failure studies could better reflect the real-life efficacy of HF therapeutics. 

### 4.5. Study Limitations

There are several limitations in our study. First, this study was a retrospective analysis, and some events could be missing if the patient was not hospitalized in our center. However, we are the only tertiary center in the geographic area, and patients listed for HT are systematically transferred in our institution in case of acute decompensation. Second, our study was focus on patients with advanced HF listed for heart transplantation, and results could not be generalizable to less severe, more co-morbid, or older heart failure patients. Third, hemodynamic variables were subject to measurement errors as well as intra- or interoperator variability. However, our center has a vast experience in pulmonary artery catheterization, limiting this bias. We did not systematically assess symptoms and clinical signs of congestion and could consequently not determine the agreement between clinical and hemodynamic assessment; however, prior studies suggest that these two approaches to congestion vary widely in advanced heart failure [37]. Next, the cohort was quite small and came from a single center. A prospective, multicentric study is needing to confirm our results and precise the best cut-off values for hemodynamic parameters. 

## 5. Conclusions

Our study highlights the feasibility and prognostic value of Forrester classification in advanced heart failure patients for stratifying the risk of events in patients awaiting heart transplant. Moreover, Forrester classification using systemic congestion is associated with the rate of composite criteria of acute heart failure, cardiogenic shock, and acute kidney injury.

## Figures and Tables

**Figure 1 jcm-11-03663-f001:**
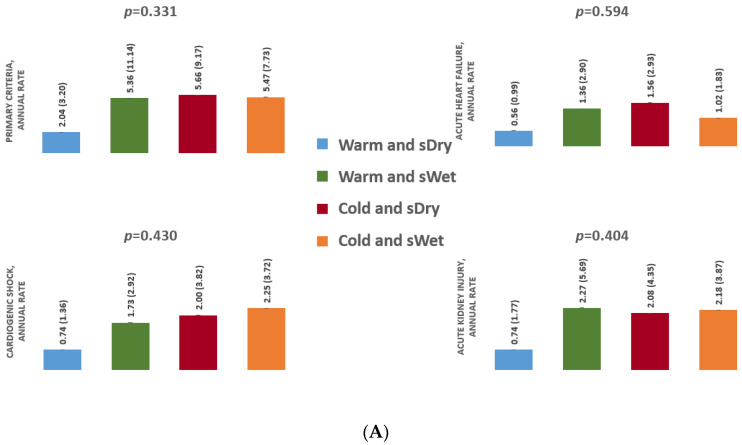
Event rate by group according to PCWP (**A**) and CVP (**B**), expressed in events/patient*year. CVP, central veinous pressure; PCWP, pulmonary capillary wedge pressure.

**Figure 2 jcm-11-03663-f002:**
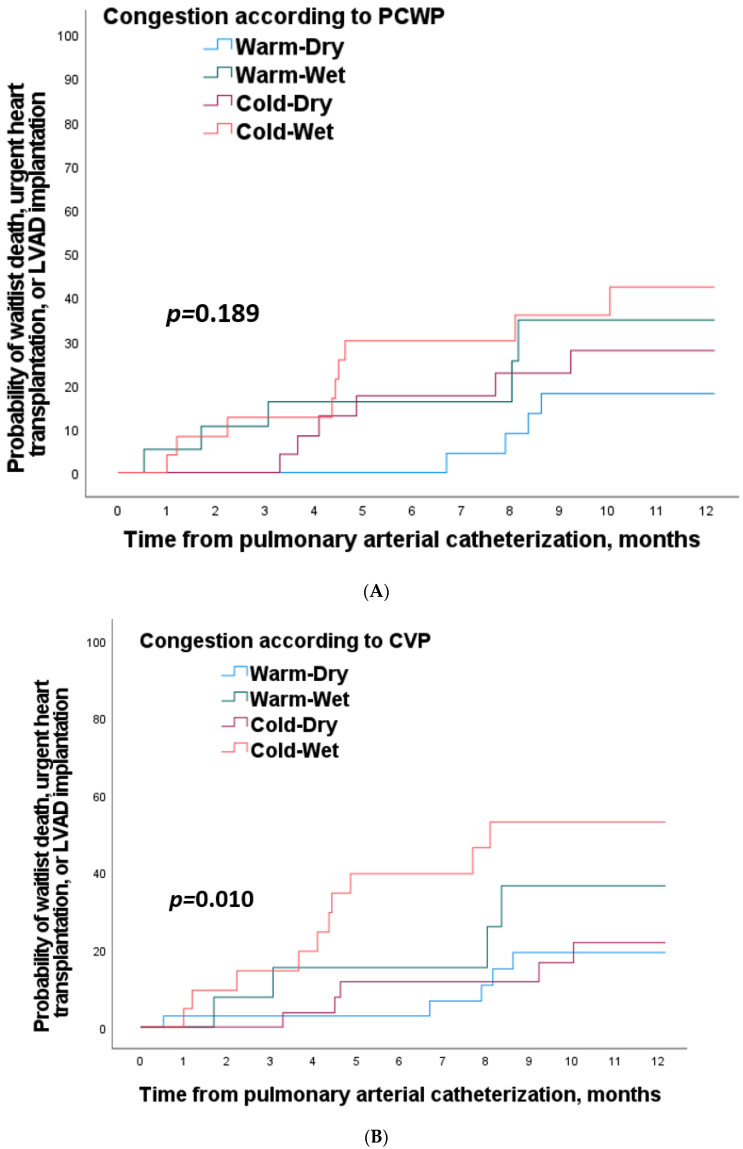
Kaplan–Meier curves for probability of waitlist death, urgent heart transplantation, or LVAD implantation by congestion according to PCWP (**A**) and CVP (**B**) at 12 months. CVP, central veinous pressure; LVAD, left ventricular assist device; PCWP, pulmonary capillary wedge pressure.

**Figure 3 jcm-11-03663-f003:**
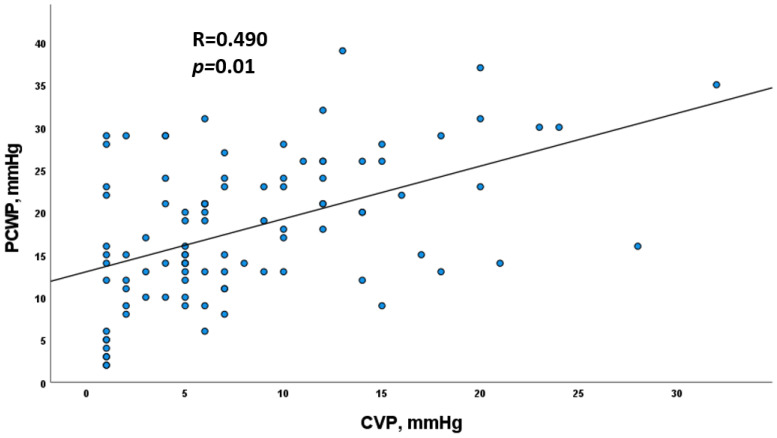
Linear correlation graph between PCWP and CVP from Pearson test. CVP, central veinous pressure; PCWP, pulmonary capillary wedge pressure.

**Table 1 jcm-11-03663-t001:** Baseline patient characteristics, medication use, and hemodynamic variables for groups according to PCWP and CVP.

(a)							
According to PCWP	Missing Data *n* (%)	Whole Cohort *n* = 100	Warm and pDry *n* = 29	Warm and pWet *n* = 19	Cold and pDry *n* = 27	Cold and pWet *n* = 25	*p*-Value
Demographic data							
Age, years	0 (0)	54 (47–61)	50 (38–60)	54 (48–61)	57 (49–62)	52 (46–59)	0.129
Male sex, *n* (%)	0 (0)	72 (72)	20 (69)	15 (79)	17 (63)	20 (80)	0.480
Body surface area, m²	0 (0)	1.91 (0.23)	1.86 (0.21)	1.88 (0.24)	1.98 (0.22)	1.96 (0.26)	0.300
**Clinical and functional parameters**							
Heart rate, beats per min	0 (0)	72 (64–85)	71 (60–90)	77 (65–86)	70 (61–79)	74 (68–86)	0.326
Systolic blood pressure, mmHg	0 (0)	101 (90–110)	105 (98–114)	95 (84–108)	93 (86–112)	101 (87–112)	0.128
Diastolic blood pressure, mmHg	0 (0)	62 (56–70)	63 (58–74)	59 (55–70)	60 (53–70)	63 (58–68)	0.454
NYHA	0 (0)	2.76 (0.53)	2.66 (0.55)	2.74 (0.65)	2.89 (0.42)	2.76 (0.52)	0.441
VO2max, mL/kg	30 (30)	12.6 (4.3)	13.9 (4.2)	14.0 (4.3)	12.1 (4.0)	11.1 (4.0)	0.152
**Medical history**							
Cardiovascular risk factors, *n* (%)							
Hypertension	0 (0)	22 (22)	4 (14)	3 (16)	8 (30)	7(28)	0.391
Diabetes mellitus	0 (0)	12 (12)	3 (10)	2 (11)	2 (7)	5 (20)	0.538
History of smoking	0 (0)	61 (61)	16 (55)	13 (68)	16 (59)	16 (64)	0.805
eGFR, *n* (%) >60 mL/min/1.73 m² <60 mL/min/1.73 m²	0 (0)	85 (85) 15 (15)	28 (97) 1 (3)	16 (84) 3 (16)	18 (66) 9 (33)	23 (92) 2 (8)	0.011
Previous cardiac surgery, *n* (%)	0 (0)	27 (27)	9 (31)	6 (32)	4 (15)	8 (32)	0.425
**Treatments and devices**							
Beta-blocker, *n* (%)	0 (0)	83 (83)	26 (90)	15 (79)	24 (89)	18 (72)	0.268
ACE inhibitor or ARB, *n* (%)	0 (0)	46 (46)	17 (59)	8 (42)	8 (29)	15 (60)	0.083
ARNI, *n* (%)	0 (0)	41 (41)	10 (34)	9 (47)	16 (59)	6 (24)	0.057
Loop diuretic dose mg	0 (0)	110 (40–160)	40 (20–125)	120 (40–160)	60 (40–125)	160 (110–250)	0.010
MRA, *n* (%)	0 (0)	82 (82)	25 (86)	17 (89)	19 (70)	21 (84)	0.307
CRT, *n* (%)	0 (0)	42 (42)	10 (34)	8 (42)	12 (44)	12 (48)	0.775
ICD, *n* (%)	0 (0)	86 (86)	25 (86)	15 (79)	26 (96)	20 (80)	0.271
**Biological data**							
Creatinine, μmol/L	0 (0)	109 (91–132)	96 (76–115)	110 (93–146)	125 (104–147)	108 (92–121)	0.017
eGFR mL/min	0 (0)	64 (46–79)	79 (53–100)	56 (44–83)	49 (38–66)	70 (55–74)	0.002
NT-proBNP, ng/L	54 (54)	2723 (974–6231)	2701 (488–3903)	3995 (2227–7962)	1909 (1171–7228)	3680 (1112–7321)	0.404
BNP, ng/L	9 (9)	744 (351–1586)	419 (137–904)	747 (381–1570)	1020 (494–2196)	1211 (591–1816)	0.014
**Echography**							
LVEDD, mm	6 (6)	67 (59–75)	67 (59–75)	68 (30–74)	66 (59–75)	65 (56–79)	0.998
LVEF, %	0 (0)	25 (20–30)	28 (24–31)	25 (20–35)	25 (20–37)	21 (18–28)	0.218
**Right heart catheterization**							
Systolic PAP, mmHg	0 (0)	40 (30–55)	29 (23–39)	55 (47–67)	31 (28–39)	56 (45–67)	<0.001
Diastolic PAP, mmHg	0 (0)	16 (12–22)	11 (6–14)	20 (18–25)	14 (10–15)	23 (20–29)	<0.001
Mean PAP, mmHg	0 (0)	26 (20–33)	21 (13–24)	33 (29–44)	21 (19–23)	34 (31–42)	<0.001
PCWP, mmHg	0 (0)	17 (12–24)	13 (6–16)	23 (21–28)	13 (11–14)	26 (22–30)	<0.001
Cardiac index, L/min/m^2^	0 (0)	2.0 (1.7–2.3)	2.4 (2.2–2.7)	2.2 (2.1–2.3)	1.7 (1.5–1.9)	1.6 (1.5–1.8)	<0.001
Cardiac out, L/min	0 (0)	3.8 (3.2–4.3)	4.5 (4.0–5.1)	4.2 (3.8–4.8)	3.3 (3.0–3.8)	3.2 (3.0–3.4)	<0.001
Pulmonary vascular resistance, Wood unit	0 (0)	2.3 (1.5–3.5)	1.5 (1.0–2.1)	2.5 (1.5–3.5)	2.5 (2.0–3.4)	2.8 (2.2–4.3)	<0.001
CVP, mmHg	0 (0)	6 (3–12)	5 (1–7)	9 (6–14)	5 (3–9)	12 (6–16)	0.003
CVP/PCWP	0 (0)	0.4 (0.3–0.6)	0.4 (0.2–0.6)	0.4 (0.3–0.6)	0.4 (0.3–0.6)	0.4 (0.2–0.6)	0.456
RVSWI	0 (0)	9.7 (7.0–12.3)	8.5 (6.8–10.5)	14.7 (11.9–16.7)	7.0 (6.0–8.8)	10.8 (9.0–12.9)	<0.001
PAPi	0 (0)	3.7 (2.2–9.5)	5.4 (2.7–13.0)	3.4 (2.2–9.7)	3.2 (2.5–8.0)	2.6 (1.7–6.7)	0.351
**(b)**							
**According to CVP**	**Missing data** ***n* (%)**	**Whole Cohort** ***n* = 100**	**Warm and sDry** ***n* = 35**	**Warm and sWet** ***n* = 13**	**Cold and sDry** ***n* = 31**	**Cold and sWet** ***n* = 21**	***p*-Value**
**Demographic data**							
Age, years	0 (0)	54 (47–61)	52 (45–60)	60 (43–62)	56 (49–61)	52(45–61)	0.483
Male sex, *n* (%)	0 (0)	72 (72)	26 (74)	9 (69)	22 (71)	15 (71)	0.984
Body surface area, m²	0 (0)	1.91 (0.23)	1.84 (0.20)	1.93 (0.25)	1.95 (0.23)	1.94 (0.25)	0.192
**Clinical and functional parameters**							
Heart rate, beats per min	0 (0)	72 (64–85)	69 (60–86)	81 (73–94)	67 (63–78)	80 (71–87)	0.027
Systolic blood pressure, mmHg	0 (0)	101 (90–110)	101 (92–110)	108 (85–112)	105 (88–116)	92 (85–105)	0.497
Diastolic blood pressure, mmHg	0 (0)	62 (56–70)	63 (57–71)	61 (53–72)	62 (58–71)	62 (53–67)	0.635
NYHA	0 (0)	2.76 (0.53)	2.57 (0.55)	3 (0.58)	2.81 (0.48)	2.86 (0.48)	0.045
VO2max, mL/kg	30 (30)	12.6 (4.3)	14 (4.3)	11.8 (3.3)	12.1 (4.2)	11.0 (4.1)	0.044
**Medical history**							
Cardiovascular risk factors, n (%)							
Hypertension	0 (0)	22 (22)	4 (11)	3 (23)	11 (35)	4 (19)	0.128
Diabetes mellitus	0 (0)	12 (12)	2 (6)	3 (23)	4 (13)	3 (14)	0.400
History of smoking	0 (0)	61 (61)	21 (60)	8 (62)	23 (74)	9 (43)	0.158
GFR, *n* (%) >60 mL/min/1.73 m² <60 mL/min/1.73 m²	0 (0)	85 (85) 15 (15)	34 (97) 1 (3)	10 (77) 3 (23)	25 (81) 6 (19)	16 (76) 5 (24)	0.092
Previous cardiac surgery, *n* (%)	0 (0)	27 (27)	7 (20)	8 (62)	5 (16)	7 (33)	0.012
**Treatments and devices**							
Beta-blocker, *n* (%)	0 (0)	83 (83)	31 (89)	10 (77)	28 (90)	14 (66)	0.100
ACE inhibitor or ARB, *n* (%)	0 (0)	46 (46)	18 (51)	7 (54)	11 (35)	12 (57)	0.393
ARNI, *n* (%)	0 (0)	41 (41)	16 (46)	3 (23)	18 (31)	4 (19)	0.019
Loop diuretic dose, mg	0 (0)	110 (40–160)	40 (20–125)	125 (90–330)	100 (40–160)	125 (90–205)	0.058
MRA, *n* (%)	0 (0)	82 (82)	31 (89)	11 (85)	23 (74)	17 (81)	0.497
CRT, *n* (%)	0 (0)	42 (42)	14 (40)	4 (31)	16 (52)	8 (28)	0.565
ICD, *n* (%)	0 (0)	86 (86)	33 (94)	7 (54)	30(97)	16 (76)	<0.001
**Biological data**							
Creatinine, μmol/L	0 (0)	109 (91–132)	97 (81–113)	131 (89–163)	112 (99–142)	113 (104–136)	0.037
GFR mL/min	0 (0)	64 (46–79)	77 (53–89)	50 (38–90)	58 (42–73)	59 (43–73)	0.013
NT-proBNP, ng/L	54 (54)	2723 (974–6231)	2418 (470–4075)	4090(3200–11529)	1856 (973–6118)	7071 (4829–8135)	0.040
BNP, ng/L	9 (9)	744 (351–1586)	555 (185–919)	902 (241–1491)	744 (417–1475)	1586 (730–2514)	0.209
**Echography**							
LVEDD, mm	6 (6)	67 (59–75)	68 (60–75)	62 (53–71)	68 (58–80)	64 (53–78)	0.550
LVEF, %	0 (0)	25 (20–30)	27 (23–30)	26 (21–50)	26 (24–31)	20(17–27)	0.156
**Right heart catheterization**							
Systolic PAP, mmHg	0 (0)	40 (30–55)	38 (25–47)	55 (34–66)	39 (31–48)	49 (34–62)	0.004
Diastolic PAP, mmHg	0 (0)	16 (12–22)	12 (8–19)	20 (17–27)	15 (12–21)	22 (15–32)	<0.001
Mean PAP, mmHg	0 (0)	26 (20–33)	22 (13–29)	33 (25–45)	23 (20–31)	33 (24–43)	<0.001
PCWP, mmHg	0 (0)	17 (12–24)	15 (8–21)	23 (18–27)	14 (12–21)	24 (16.5–30)	<0.001
Cardiac index, L/min/m^2^	0 (0)	2.0 (1.7–2.3)	2.3 (2.1–2.7)	2.2 (2.1–2.5)	1.7 (1.6–1.9)	1.6 (1.4–1.8)	<0.001
Cardiac out, L/min	0 (0)	3.8 (3.2–4.3)	4.2 (4.0–4.7)	4.7 (3.8–5.2)	3.3 (3.0–3.8)	3.1 (2.7–3.2)	<0.001
Pulmonary vascular resistance, Wood unit	0 (0)	2.3 (1.5–3.5)	1.6 (1.1–2.6)	2.3 (1.5–3.3)	2.6 (2.0–3.6)	2.5 (2.0–4.3)	0.004
CVP, mmHg	0 (0)	6 (3–12)	5 (1–6)	14 (11–20)	5 (2–6)	14 (12–18)	<0.001
CVP/PCWP	0 (0)	0.4 (0.3–0.6)	0.3 (0.2–0.4)	0.6 (0.5–0.8)	0.3 (0.2–0.4)	0.6 (0.5–0.9)	<0.001
RVSWI	0 (0)	9.7 (7.0–12.3)	10.1 (7.2–12.9)	15.2 (11.2–16.7)	8.4 (6.7–10.7)	9.4 (6.0–11.7)	<0.001
PAPi	0 (0)	3.7 (2.2–9.5)	6.7 (4.0–13.5)	2.0 (1.4–3.3)	5.8 (3.2–12.3)	1.7 (1.0–2.5)	<0.001

ACE, angiotensin-converting enzyme; ARB, angiotensin II receptor blocker; ARNI, angiotensin receptor-neprilysin inhibitor; BNP, B-type natriuretic peptide; Ci, cardiac index; CRT, cardiac resynchronization therapy; CVP, central veinous pressure; eGFR, estimated glomerular filtration rate; ICD, implantable cardioverter-defibrillator; LVEDD, left ventricular end-diastolic diameter; LVEF, left ventricular ejection fraction; MRA, mineralocorticoid receptor antagonist; NYHA, New York Heart Association; PAP, pulmonary artery pressure; PAPi, pulmonary artery pressure index; PCWP, pulmonary capillary wedge pressure; CVP/PCWP, central veinous pressure divided by pulmonary capillary wedge pressure; RVSWI, right ventricular stroke work index; VO2max, maximum oxygen consumption rate.

## Data Availability

Not applicable.

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
