# Peer review of "Prognosis of Advanced Heart Failure Patients according to Their Hemodynamic Profile Based on the Modified Forrester Classification"

_jcm, 2022, doi:10.3390/jcm11133663_

Round 1
Reviewer 1 Report
Despite the interesting topic and results of this manuscript by Baudry et al., I have one major concern that should be addressed by the authors:
According to the baseline characteristics, there are some significant differences between the four patient groups. For instance, the use of ARNIs in ‘cold and swet’ patients appears to be rather low compared to ‘warm and sdry’ patients. Moreover, the percentage of impaired kidney function (GFR <60 ml/min/1.73m2 in ‘cold and swet’ was 24 %, while it was only 3 % in ‘warm and sdry’ patients. These differences could have had a significant impact on the primary and secondary endpoints. A multivariate analysis seems more appropriate to evaluate whether the Forrester classification serves as an independent predictor of cardiorenal outcomes.
Author Response
Reviewer 1 :
Despite the interesting topic and results of this manuscript by Baudry et al., I have one major concern that should be addressed by the authors:
According to the baseline characteristics, there are some significant differences between the four patient groups. For instance, the use of ARNIs in ‘cold and swet’ patients appears to be rather low compared to ‘warm and sdry’ patients. Moreover, the percentage of impaired kidney function (GFR <60 ml/min/1.73m2 in ‘cold and swet’ was 24 %, while it was only 3 % in ‘warm and sdry’ patients. These differences could have had a significant impact on the primary and secondary endpoints. A multivariate analysis seems more appropriate to evaluate whether the Forrester classification serves as an independent predictor of cardiorenal outcomes.
We agree with this reviewer’s concern regarding the difference in baseline characteristics between groups. We added a sentence in the manuscript to emphasize this important aspect as follows:
“Whether based on PCWP or CVP, eGFR was significantly different between group (p=0.002 according to PCWP and p=0.013 according to CVP). Loop diuretic dose was significantly different between Forrester’s profile according to PCWP (p=0.010) but only tended to differe when CVP was considered (p=0.058). Other treatments were comparable between groups excepted ARNI: Patients with systemic congestion had a lower likelihood of receiving ARNI (p=0.019).”
We performed a Cox proportional hazard regression to evaluate the value of Forrester’s classification according to CVP and PCWP. The results were overall unchanged. We have added the following tables in the supplementary material and modified text accordingly:
Analysis according to pulmonary congestion
The percentage of waitlist death, emergency HT or LVAD implantation at 12 months was 36.0%, 26.3%, 22.2% and 13.2% in “Cold & pWet”, “Warm & pWet”, “Cold & pDry” and “Warm & pDry” profiles, respectively (p=0.189) (Figure 2a). Forrester’s profile according to pulmonary congestion were not significantly associated with secondary endpoints in multivariate analysis (supplemental table 1).
Analysis according to systemic congestion
The percentage of waitlist death, emergency HT or LVAD implantation at 12 months was 47.6%, 30.8%, 16.1% and 14.3% in “Cold & sWet”, “Warm & sWet”, “Cold & sDry” and “Warm & sDry” profiles, respectively (p=0.010) (Figure 2b). The “Cold-Wet” profile was associated with a 240% increase in this secondary outcome while adjusting on age, gender and eGFR at RHC (HR, 3.40, 95%CI, 1.12 to 10.29; p=0.030) (supplemental table 1).
|
Model adjusted on age, gender and eGFR at RHC. |
||||
|
According to CVP |
Warm-Dry |
Cold-Dry |
Warm-Wet |
Cold-Wet |
|
Waitlist death or urgent heart transplant or LVAD implantation |
Reference |
1.79 (0.46 - 6.93) p=0.399
|
0.88 (0.25 - 3.14) p=0.842
|
3.40 (1.12 - 10.29) p=0.030 |
|
According to PCWP |
Warm-Dry |
Cold-Dry |
Warm-Wet |
Cold-Wet |
|
Waitlist death or urgent heart transplant or LVAD implantation |
Reference |
2.00 (0.53 - 7.57) p=0.308
|
1.25 (0.33 - 4.67) p=0.742
|
2.82 (0.86 - 9.24) p=0.087 |

Reviewer 2 Report
The paper of Baudry and coll. is very interesting. However, the authors should consider some points.
- The authors do not report the symptoms and the clincal signs useful to assess the presence of congestion and/or low cardiac output. This is a very relevant issue. The agreement between Forrester and clinical classification could be very useful for readers.
- No data about pulmonary hypertension reversibility were reported. This could be even more relevant when patients’ outcome is considered
- The events should be reported in details as total number observed. The event rate should be better clarified. Do the event rate refer to the number of events for patients for year?
- Number of deaths and cause of deaths should be reported.
- Figure 4 is not clear. It could be improved.
Minor points:
- References should be reordered according with the citation throughout the text
- Authors refers to 2016 Guidelines. They should be updated.
- Further details about the French national CRISTAL database should be provided.
Author Response
Reviewer 2:
The paper of Baudry and coll. is very interesting. However, the authors should consider some points.
- The authors do not report the symptoms and the clinical signs useful to assess the presence of congestion and/or low cardiac output. This is a very relevant issue. The agreement between Forrester and clinical classification could be very useful for readers.
We do agree that symptoms and clinical signs would be useful to assess the agreement between Forrester hemodynamic and clinical classification. Indeed, important differences between clinical and hemodynamic evaluation have been reported in a number of advanced or acute HF studies (Shah MR, Hasselblad V et al. J Card Fail 2001;7:105–113 & Narang N, Chung B et al. J Card Fail Elsevier Inc.; 2020;26:128–135).
Unfortunately, we do not have a systematic assessment of clinical congestion. This is a limitation that we included in the revised version fo the manuscript as follows:
We did not systematically assess symptoms and clinical signs of congestion and could consequently not determine the agreement between clinical and hemodynamic assessment; however, prior studies suggest that these 2 approaches to congestion vary widely in advanced heart failure.
- No data about pulmonary hypertension reversibility were reported. This could be even more relevant when patients’ outcome is considered
Fixed pulmonary arterial hypertension is a contraindication to heart transplant listing according to 2016 ISHLT guidelines. As a consequence, none of the patients whose data is reported herein had fixed pulmonary arterial hypertension.
We modified the “Pulmonary artery catheterization” paragraph to highlight this important point as follow:
Right heart catheterization measurements were obtained as part of the assessment for heart transplant listing according to the 2016 International Society for Heart Lung Transplantation guidelines (Class 1, Level of Evidence: C). Elevated pulmonary vascular resistance (PVR) that is refractory to medical therapy may be a contraindication to HT.
- The events should be reported in details as total number observed. The event rate should be better clarified. Do the event rate refer to the number of events for patients for year?
We used an event rate that is indeed measured according to patient*year. We have clarified this point by modifying the manuscript:
The primary composite endpoint was the annual rate of cardiogenic shock, acute kidney injury and acute heart failure. Rates were the mean of the number of events during the waiting period on heart transplantation list divided by the time spent on waitlist in years in each patient and were expressed in events/patient*year .
- Number of deaths and cause of deaths should be reported.
We thank the reviewer and modified the manuscript as follows:
Mortality
10 patients died on the waiting list, (6 patients related to cardiomyopathy, 2 from a pathology unrelated to heart disease and 2 of unknown cause).
- Figure 4 is not clear. It could be improved.
We agree with the reviewers that Figure 4 is confusing and we have decided to remove it.
Minor points:
- References should be reordered according with the citation throughout the text
We thank the reviewer and have modified the manuscript accordingly.
- Authors refers to 2016 Guidelines. They should be updated.
We thank the reviewer and have modified the manuscript accordingly.
- Further details about the French national CRISTAL database should be provided.
CRISTAL is a national database initiated in 1996 and administered by the Agence de la biomédecine that prospectively collects data on all organ transplant candidates in France along with their outcomes. Data are entered into the registry by each center. Data collection is mandatory.
We upgraded the description of the Cristal database in the “data collection” paragraph as follows: This single-center cohort was performed using the data of the French national transplant registry (CRISTAL). This registry is managed by the Agence de la biomédecine and prospectively collected data from every candidate for organ transplant in France. Data collection is required by the transplant agency.

Round 2
Reviewer 2 Report
The authors answered to the questions raised. I've no further comment.